# Characterization of Respiratory Viruses in Patients with Acute Respiratory Infection in the City of Barranquilla during the SARS-CoV-2/COVID-19 Pandemic

**DOI:** 10.3390/diagnostics14202269

**Published:** 2024-10-12

**Authors:** Leonardo Arrieta-Rangel, Yesit Bello-Lemus, Ibeth Luna-Rodriguez, Martha Guerra-Simanca, Valmore Bermúdez, Yirys Díaz-Olmos, Elkin Navarro Quiroz, Lisandro Pacheco-Lugo, Antonio J. Acosta-Hoyos

**Affiliations:** 1Life Sciences Research Center, Universidad Simón Bolívar, Barranquilla 080002, Colombia; leonardo.arrieta@unisimon.edu.co (L.A.-R.); yesit.bello@unisimon.edu.co (Y.B.-L.); ibeth.luna@unisimon.edu.co (I.L.-R.); martha.guerra@unisimon.edu.co (M.G.-S.); elkin.navarro@unisimon.edu.co (E.N.Q.); lisandro.pacheco@unisimon.edu.co (L.P.-L.); 2School of Health Sciences, Universidad Simón Bolívar, Barranquilla 080002, Colombia; valmore.bermudez@unisimon.edu.co; 3Health Sciences Division, Universidad del Norte, Barranquilla 080003, Colombia; olmosy@uninorte.edu.co

**Keywords:** influenza-like illness, acute respiratory infection, molecular diagnosis, RT-qPCR, epidemiological surveillance, public health, quantitative polymerase chain reaction

## Abstract

**Introduction:** Severe acute respiratory infection (SARI) is mainly caused by viral pathogens, with a high prevalence in high-risk populations such as infants and older adults. Coinfections by different viruses are frequent and, in some cases, associated with severe disease outcomes. **Purpose:** The main purpose of this study was to identify respiratory viruses circulating in Barranquilla during the peaks of the COVID-19 pandemic and estimate the prevalence of viral coinfections in samples from individuals with different degrees of respiratory infection. **Methods:** We received 5083 samples between epidemiological weeks 33–42 of 2021 submitted by the District Health Laboratory of Barranquilla and four local healthcare institutions during COVID-19 surveillance. Among them, we analyzed 101 samples from individuals presenting with influenza-like illness (ILI). Eighteen respiratory viruses, including SARS-CoV-2, were evaluated via qRT-PCR using nasal swabs or nasopharyngeal aspirate samples. **Results**: Of the 101 study individuals, 56 were male and 45 were female (55.5% and 44.5%, respectively); 25.7% of individuals were infected with at least one of the evaluated viruses. Respiratory syncytial virus (RSV) and human rhinovirus (HRV) were the two most frequently detected viruses (30.7% and 15.4% of total positives, respectively). Coinfections with two or more respiratory viruses accounted for 42% of the total positive cases. **Discussion:** Our findings indicate the presence of different respiratory viruses in swab or nasopharyngeal aspirate samples from individuals with ILI, including coinfections. These results reveal the circulation of several respiratory viruses in the city of Barranquilla, confirming their importance as potential causes of SARI in Colombia and the need for their active surveillance.

## 1. Introduction

Acute respiratory infection (ARI) is a major public health concern worldwide and is defined as “an infectious process of the upper and lower respiratory tracts caused by bacteria and viruses, with progression of less than 15 days” [1]. However, it is estimated that in 80% of cases, the etiological agent is viral [1]. The most frequent forms of transmission are direct contact with infected subjects and contact with fomites [2]. ARI mainly affects children under 5 years of age, adults over 60 years of age, individuals with chronic underlying diseases, and immunosuppressed patients [3].

In Colombia, surveillance of this disease is framed within four strategies: the sentinel surveillance of influenza-like illness-severe acute respiratory infection (ILI-SARI), the intensified surveillance of unusual SARI cases, the surveillance of ARI mortality in children under 5 years of age, and the surveillance of ARI morbidity in all age groups, encompassing both hospitalized and ambulatory patients [3]. These strategies focus on establishing risk factors and directing promotion, prevention, and control measures.

Local longitudinal active sampling is an alternative to determining the ARI trend [4]. Over the last few years, the ability to detect respiratory viruses has improved owing to polymerase chain reaction (PCR) tests and their quantitative variants (qPCR and RT-qPCR). As a result, the spectrum of respiratory viruses detected through surveillance actions has broadened [5]. Monitoring circulating respiratory viruses can help to identify patterns in respiratory infections in Barranquilla, contributing to the development of strategies to prevent outbreaks, epidemics, and, eventually, pandemics.

Colombia, at the beginning of the COVID-19 pandemic in 2020, similar to many other countries around the world, implemented measures to mitigate SARS-CoV-2’s propagation and alleviate the burden on the population. Measures including handwashing, social distancing, and the use of face masks were mandatory in all social interactions and proved an effective way to combat the disease. However, even with these restrictions, from the beginning of the pandemic to the end of 2021, worldwide, Colombia ranked 13th based on the number of COVID-19 cases reported and 11th based on the number of deaths (see WHO: https://www.who.int/docs/default-source/coronaviruse/situation-reports/20200326-sitrep-66-covid-19.pdf (accessed on 4 September 2024)). In this scenario, the attention of ARI researchers was directed towards SARS-CoV-2 infection, while reports of flu and ORV were significantly low. During 2021, the number of ILI cases submitted by sentinel surveillance was 0.02% of the number of cases investigated for SARS-CoV-2 (3.440 ILI/12.189.576 COVID-19). Using the RT-qPCR technique, our study aimed to identify the respiratory viruses causing ARI-SARI circulating in Barranquilla between August and October 2021, during the COVID-19 pandemic, and to estimate the prevalence of viral coinfections in samples from individuals with different degrees of respiratory infection.

## 2. Materials and Methods

### 2.1. Sample Reception

The samples used to detect respiratory viruses were provided by the District Health Laboratory of Barranquilla and four local healthcare institutions. The samples were sent to the Molecular Diagnostics Laboratory of Universidad Simón Bolívar in triple packaging and under adequate refrigeration conditions (2–8 °C). For patients undergoing outpatient and emergency care, nasopharyngeal swab (NPS) samples were submitted, whereas for patients undergoing hospitalized and intensive care, nasopharyngeal aspirates (NPA) were used. The NPS samples were sent in a viral transport medium and the NPA samples were collected in a sterile saline solution by a trained healthcare professional in a volume of no less than 2 mL, in accordance with the guidelines of Colombia’s National Institute of Health (INS) [6].

### 2.2. Selection Criteria

In this study, 5083 individuals whose samples were received during epidemiological weeks 33–42 of 2021 (August–October) were included (Figure 1). For the surveillance process, 101 typical symptomatic ILI cases were selected. Samples for the detection of respiratory viruses were collected from cases with acute episodes of fever, headache, and coughing with no more than 7 days of progression, along with symptoms such as adynamia, odynophagia, rhinorrhea, and respiratory distress. Similarly, patients with pre-existing diseases as a predisposing factor were included in the selection profile. Accordingly, SARS-CoV-2 infection was also used for the subsequent grouping of the cases, wherein 36 were positive and 65 were negative for the presence of this virus. The primary source of information was the data provided in the patient’s discharge report and epidemiological record (FE 345-346-348, INS) for each case.

### 2.3. Total RNA/DNA Extraction and RT-qPCR for Respiratory Viruses

RNA/DNA extraction was performed using the commercial Quick-DNA/RNA Viral MagBead kit (Zymo Research, Irvine, CA, USA) following the manufacturer’s instructions. Briefly, 200 µL aliquots of each sample (NPA or NPS) were treated with the lysis buffer provided in the kit. Subsequently, the RNA and DNA molecules released using the previous procedure were adsorbed on magnetic beads and subjected to a series of washes with different buffer solutions and ethanol. Finally, total RNA/DNA was eluted in 50 µL of nuclease-free water.

The detection of respiratory viruses was performed via RT-PCR using the Allplex™ Respiratory Panel 1, Respiratory Panel 2, and Respiratory Panel 3 kits (Seegene Inc, Seoul, South Korea) (Table 1).

The reactions were prepared according to the manufacturer’s instructions, using a total volume of 25 µL per reaction: 17 µL of reaction mix and 8 µL of RNA/DNA from each sample. The amplification reactions were performed on the CFX96™ Real-time PCR System thermal cycler (Bio-Rad, Hercules, CA, USA).

## 3. Data Analysis

The collected information was tabulated and analyzed using descriptive statistical tools. For this purpose, a database was created in Excel™ format (Microsoft^®^ Excel^®^ for Microsoft 365 MSO version 2408), in which the relevant variables of the patients evaluated were recorded: sex, age, severity criteria, comorbidities (if any), and the symptoms described in the epidemiological records. Contingency tables and independence tests were estimated using the OpenEpi Version 3 platform. Using RT-qPCR tests, the prevalence of respiratory viruses during each epidemiological week was established.

## 4. Results

Our study reports the circulation of a group of respiratory viruses commonly associated with ARI in the city of Barranquilla. These viruses include respiratory syncytial virus (RSV A and B), parainfluenza virus (PIV 2, 3, and 4), human rhinovirus (HRV), metapneumovirus (MPV), adenovirus (AdV), human enterovirus (HEV), and three coronavirus subtypes (CoV OC43, CoV 229E, and NL63) between epidemiological weeks 33 and 42 of 2021. Using qRT-PCR, a prevalence of 36.9% was observed in patients infected with one or more respiratory virus other than SARS-CoV-2 and 5.5% for those infected with SARS-CoV-2 (Table 2 and Table 3, respectively). Interweek positivity showed an increasing trend. Of the included patients, 58.4% were hospitalized, 40.7% of whom required a high complexity of care. Additionally, 2% were post-mortem patients.

RSV and HRV were the most frequently detected viruses, with eight and four cases (representing 30.7% and 15.4% of the total positives, respectively). Coinfections with two or more respiratory viruses accounted for 42.3% of positive cases (Table 4).

Most study patients were infants aged 0–5 years, and this age group exhibited the greatest diversity of respiratory viruses (Table 5).

Based on the collected data, we studied the possible dependence relationship between SARS-CoV-2 infection and the symptoms caused by other respiratory viruses. Therefore, although there was a dependence between exposure to SARS-CoV-2 and the occurrence of ARI caused by other respiratory viruses (χ^2^: 10.34; *p*-value: 0.00015), we found that this association was relatively low (OR: 0.102; CI: 0.015–0.407).

Coinfection with PIV-4 and MPV was the most common and was observed in 50% of individuals with more than one viral agent detected. The highest number of combinations between respiratory viruses causing ARI-SARI occurred in infants. The frequency of coinfections was predominant among hospitalized adults (28–60 years) and older adults (60–80 years), 80% of whom already had pre-existing diseases (chronic obstructive pulmonary disease (COPD), active pneumonia, and/or tuberculosis).

Of the patients, 65.3% had no comorbidities. In the remaining patients, the following comorbidities were observed: diabetes (14.8%), pneumonia (14.8%), heart disease (7.4%), severe respiratory disease (COPD, asthma) (22.2%), autoimmune disease (7.4%), HIV (3.7%), tuberculosis (3.7%), and the involvement of other systems (7.4%). In 25.7% of these cases, one or more respiratory virus was detected (Table 6).

## 5. Discussion

Within the context of the SARS-CoV-2/COVID-19 health emergency, the prevalence of respiratory viruses was investigated among patients with ARI-SARI symptoms. The results indicate a significant variety of respiratory viruses in Barranquilla. The prevalence reported in this study aligns with findings from other studies in Colombia and Latin America [7,8]. However, the variability in the dynamics of infection caused by viral agents has been linked to the weather characteristics of the tropical zone [9].

According to INS figures, during weeks 33–42 of 2021, the sentinel surveillance for ARI-SARI identified several viruses, including RSV (55.2%), HRV (11.2%), AdV (20.8%), hMPV (5.9%), PIV (3.7%), and CoV (0.7%) [10,11,12]. The values for RSV and HRV infections were similar to those reported in this study. For PIV, the kit allowed for the segregation of different virus serotypes, and analysis was conducted accordingly. Conversely, PIV was frequently found in coinfections with one or more respiratory virus.

In 2018 and 2019, the INS reported RSV in approximately 50% of ARI cases, followed by influenza A (Flu A) and adenovirus (AdV) as the most prevalent respiratory viruses over the same epidemiological period of interest [13]. This finding is consistent with that of our study. AdV was found in low proportions and only in coinfections with other viruses. Conversely, due to the COVID-19 pandemic, sentinel surveillance of ARI-SARI was suspended during weeks 33–42 of 2020, except for week 40, when cases of RSV and AdV [3] were reported. The incidence of SARS-CoV-2 in Barranquilla was 88.7/100,000 inhabitants (CI: 71.48–105.96) during the study period, compared to 207.58/100,000 inhabitants in the Colombian territory as a whole [14].

RSV infection in children under 5 years of age has previously been reported in Colombia [7,8,15,16]. RSV incidence in patients under 1 year of age with lower tract ARI ranged from 30% to 46%, with peaks between March and May, coinciding with the first annual rainy season [8,15]. In Brazil, 17.5% positivity was found in children aged 0–12 years between 2009 and 2013 and a higher viral load was detected in hospitalized cases [16]. Likewise, our findings are consistent with previously reports from the United States on RSV ARI in this age group [17].

The frequency of coinfections (41.7%) found in our study was similar to that described in other studies [18,19] and significantly higher than that described in a study in Santander, Colombia [20] and one in Santiago, Chile [21]. Some authors suggest that viral coinfections occurring in ARI cases in children are associated with exacerbated symptoms and require longer hospital and/or ICU stays [22,23]. Despite this finding, Asner et al. and other systematic reviews indicate that the presence of two or more respiratory viruses is not associated with ARI-SARI severity or its clinical course [24,25,26]. Coinfection of hMPV with other viruses has previously been reported, and the associated agents are usually PIV, RSV, and HRV [21,23].

For the purposes of INS surveillance, influenza virus (Flu) was not detected in the samples based on the operational definition of ILI, suggesting a low prevalence. This finding aligns with another study reporting flu only in a coinfection with SARS-CoV-2 and RSV [27].

Our study had several limitations. Although RT-qPCR was used, not all samples from hospitalized patients with ARIs were screened owing to limitations in reagent availability. A larger sample size would be needed to make broader inferences about the general population. Additionally, our study captured virus behavior during a specific time period, which may have been influenced by serotypes and environmental variations, among other factors. However, according to sentinel surveillance from previous years, the timeframe chosen (epidemiological weeks 33–42) presented a high incidence of respiratory viruses [13] and also coincided with a low peak of SARS-CoV-2 in our region, which facilitated the work in our laboratory to carry out these experiments.

Despite SARS-CoV-2 being the predominant virus in ARI-SARI during the COVID-19 pandemic, our findings suggest that surveillance of other viruses as potential etiologic agents of SARI should continue. This should be supported by both the reference laboratory and the national network of laboratories. Further studies in other territories are necessary to characterize circulating viruses and develop effective prevention and promotion measures.

## Figures and Tables

**Figure 1 diagnostics-14-02269-f001:**
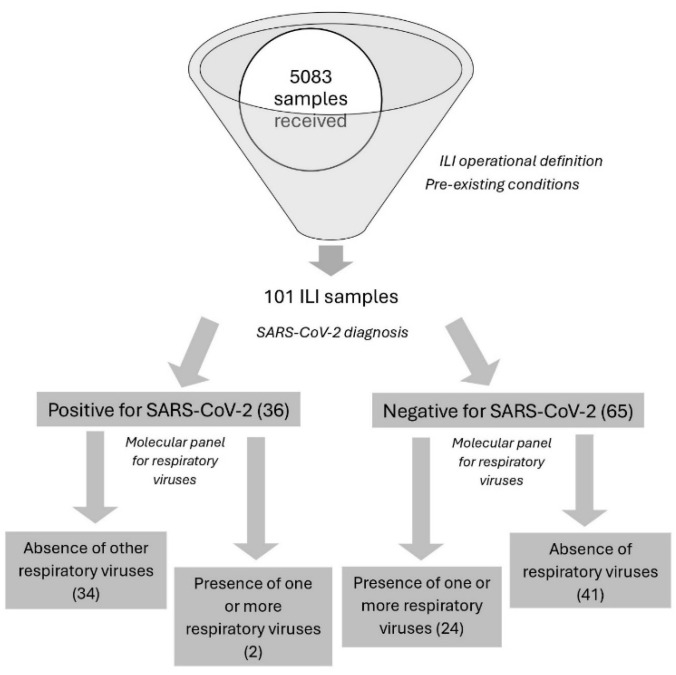
Flow chart used for the selection and screening of respiratory samples.

**Table 1 diagnostics-14-02269-t001:** Respiratory viruses screened in this study.

Allplex™ Respiratory Panel 1	Allplex™ Respiratory Panel 2	Allplex™ Respiratory Panel 3
Influenza A (Flu A)	Human adenovirus (AdV)	Bocavirus 1/2/3/4 (BoV)
Influenza B (Flu B)	Human metapneumovirus (hMPV)	Human rhinovirus A/B/C (HRV)
Respiratory Syncytial Virus A (RSV A)	Human enterovirus (hEV)	Coronavirus 229E
Respiratory Syncytial Virus B (RSV B)	Parainfluenza 1 (PIV-1)	Coronavirus NL63
Influenza A-H1N1 (Flu A-H1pdm09)	Parainfluenza 2 (PIV-2)	Coronavirus OC43
	Parainfluenza 3 (PIV-3)	
	Parainfluenza 4 (PIV-4)	

**Table 2 diagnostics-14-02269-t002:** Sociodemographic variables and comorbidities among ILI patients who were positive for SARS-CoV-2.

Variables	Positive SARS-CoV-2 Diagnosis							
	No Hospitalization		Hospitalization		Intensive Care Unit Admission		Deaths	
	n	%	n	%	n	%	n	%
**Sex**								
Male	6	5.94%	1	0.99%	1	0.99%	-	-
Female	13	12.87%	11	10.89%	2	1.98%	2	1.98%
**Age (years)**								
0–5	1	0.99%	-	-	-	-	-	-
6–11	-	-	-	-	-	-	-	-
12–18	1	0.99%	-	-	-	-	-	-
19–27	8	7.92%	-	-	-	-	-	-
28–59	7	6.93%	3	2.97%	2	1.98%	2	1.98%
60–80	2	1.98%	8	7.92%	1	0.99%	-	-
>80	-	-	1	0.99%	-	-	-	-
**Comorbidities**								
Autoimmune disease	-	-	-	-	-	-	-	-
Diabetes	-	-	-	-	-	-	-	-
Immunosuppressed individuals	1	0.99%	-	-	-	-	-	-
Digestive system diseases	1	0.99%	-	-	-	-	-	-
Cardiovascular system diseases	-	-	2	1.98%	-	-	-	-
Nervous system diseases	-	-	-	-	-	-	-	-
Respiratory system diseases	1	0.99%	1	0.99%	1	0.99%	1	0.99%
Urinary system diseases	-	-	-	-	-	-	-	-
Various comorbidities	1	0.99%	-	-	1	0.99%	1	0.99%
Human Immunodeficiency virus	-	-	-	-	-	-	-	-
None	15	14.85%	9	8.91%	1	0.99%	-	-
**Molecular diagnosis (respiratory panels)**								
Negative	17	16.83%	12	11.88%	3	2.97%	2	1.98%
Positive	2	1.98%	-	-	-	-	-	-

**Table 3 diagnostics-14-02269-t003:** Sociodemographic variables and comorbidities among ILI patients who were negative for SARS-CoV-2.

Variables	Negative SARS-CoV-2 Diagnosis					
	No Hospitalization		Hospitalization		Intensive Care Unit Admission	
	n	%	n	%	n	%
**Sex**						
Male	13	20.0%	12	18.5%	12	18.5%
Female	8	12.3%	11	16.9%	9	13.8%
**Age groups (years)**						
0–5	5	7.7%	11	16.9%	10	15.4%
6–11	6	9.2%	1	1.5%	1	1.5%
12–18	3	4.6%	1	1.5%	1	1.5%
19–27	1	1.5%	1	1.5%	2	3.1%
28–59	3	4.6%	4	6.2%	5	7.7%
60–80	2	3.1%	3	4.6%	2	3.1%
>80	1	1.5%	2	3.1%	-	-
**Comorbidities**						
Autoimmune disease	-	-	-	-	1	1.5%
Diabetes	2	3.1%	-	-	-	-
Immunosuppressed individuals	1	1.5%	1	1.5%	1	1.5%
Cardiovascular system diseases	2	3.1%			1	1.5%
Nervous system diseases	-	-	1	1.5%	1	1.5%
Respiratory system diseases	1	1.5%	3	4.6%	3	4.6%
Urinary system diseases	-	-	-	-	1	1.5%
Human Immunodeficiency virus	-	-	1	1.5%	-	-
Various comorbidities	-	-	3	4.6%	1	1.5%
Autoimmune disease	-	-	-	-	1	1.5%
None	15	23.1%	14	21.5%	12	18.5%
**Molecular diagnosis (respiratory panels)**						
Negative	19	29.2%	9	13.8%	13	20.0%
Positive	2	3.1%	14	21.5%	8	12.3%

**Table 4 diagnostics-14-02269-t004:** Distribution of respiratory viruses found in this study.

Etiological Agent	n	%
**RSV A**	**4**	**15.38%**
**RSV B**	**4**	**15.38%**
**PIV 2**	**1**	**3.85%**
**PIV 3**	**2**	**7.69%**
**HRV**	**4**	**15.38%**
**>1 respiratory virus**	**11**	**42.31%**
AdV, HEV, HRV	1	3.85%
RSV A, AdV, HEV	1	3.85%
RSV B, HRV	1	3.85%
PIV 2, HRV	1	3.85%
PIV 4, MPV	4	15.38%
PIV 4, MPV, HRV	1	3.85%
HRV, Coronavirus subtype 229E, Coronavirus subtype NL63	1	3.85%
HRV, Coronavirus subtype OC43	1	3.85%
**Total**	**26**	

Note: All abbreviations referred to in this table stand for the viruses studied as follows: RSV: respiratory syncytial virus; PIV: parainfluenza virus; HRV: human rhinovirus; AdV: adenovirus; HEV: human enterovirus; MPV: human metapneumovirus.

**Table 5 diagnostics-14-02269-t005:** Respiratory virus distribution by age group.

Etiological Agent	SARS-CoV-2 Negative													SARS-CoV-2 Positive				
	0–5		6–11		19–27		28–59		60–80		>80		Total	19–27		28–59		Total
	n	%	n	%	n	%	n	%	n	%	n	%		n	%	n	%	
**RSV A**	**3**	**11.54**	-	-	**1**	**3.85**	-	-	-	-	-	-	**4**	-	-	-	-	
**RSV B**	**3**	**11.54**	**1**	**3.85**	-	-	-	-	-	-	-	-	**4**	-	-	-	-	
**PIV 2**	**1**	**3.85**	-	-	-	-	-	-	-	-	-	-	**1**	-	-	-	-	
**PIV 3**	**1**	**3.85**	-	-	-	-	-	-	-	-	1	3.85	**2**	-	-	-	-	
**HRV**	**1**	**3.85**	-	-	**1**	**3.85**	-	-	**1**	**3.85**	-	-	**3**	-	-	**1**	**3.85**	**1**
**>1 respiratory virus**	**6**	**23.08**	-	-	-	-	**3**	**11.54**	**1**	**3.85**	-	-	**10**	-	-	-	-	
AdV, HEV, HRV	1	3.85	-	-	-	-	-	-	-	-	-	-	1	-	-	-	-	
RSV A, AdV, HEV	1	3.85	-	-	-	-	-	-	-	-	-	-	1	-	-	-	-	
RSV B, HRV	-	-	-	-	-	-	1	3.85	-	-	-	-	1	-	-	-	-	
PIV 2, HRV	1	3.85	-	-	-	-	-	-	-	-	-	-	1	-	-	-	-	
PIV 4, MPV	2	7.69	-	-	-	-	1	3.85	1	3.85	-	-	4	-	-	-	-	
PIV 4, MPV, HRV	1	3.85	-	-	-	-	-	-	-	-	-	-	1	-	-	-	-	
HRV, Coronavirus subtype 229E, Coronavirus subtype 229E	-	-	-	-	-	-	-	-	-	-	-	-	-	**1**	**3.85**	-	-	1
HRV, Coronavirus subtype OC43	-		-	-	-	-	1	3.85	-	-	-	-	1	-	-	-	-	
													**24**					**2**

Note: All abbreviations referred to in this table stand for the viruses studied as follows: RSV: respiratory syncytial virus; PIV: parainfluenza virus; HRV: human rhinovirus; AdV: adenovirus; HEV: human enterovirus; MPV: human metapneumovirus.

**Table 6 diagnostics-14-02269-t006:** Respiratory virus distribution in patients with comorbidities.

Comorbidity/Etiological Agent	No Hospitalization		General Hospitalization		Intensive Care Unit Admission		Total
	n	%	n	%	n	%	
**Immunosuppressed**							**1**
RSV A	-	-	-	-	1	3.8	
**Nervous system diseases**							**1**
PIV 2	-	-	1	3.8	-	-	
**Respiratory system diseases**							**4**
PIV 4, MPV	-	-	-	-	1	3.8	
HRV, Coronavirus subtype OC43	-	-	-	-	1	3.8	
RSV B	-	-	1	3.8	-	-	
HRV	-	-	1	3.8	-	-	
**Urinary system diseases**							**1**
HRV	-	-	-	-	1	3.8	
**Various comorbidities**							**2**
PIV 4, MPV	-	-	1	3.8	-	-	
PIV 3	-	-	1	3.8	-	-	
**None**	4	**15.4**	**9**	**34.6**	**4**	**15.4**	**17**
							**24**

Note: All abbreviations referred to in this table stand for the viruses studied as follows: RSV: respiratory syncytial virus; PIV: parainfluenza virus; HRV: human rhinovirus; MPV: human metapneumovirus.

## Data Availability

All data presented here is available upon request to authors.

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
