# Peer review of "Characterization of Respiratory Viruses in Patients with Acute Respiratory Infection in the City of Barranquilla during the SARS-CoV-2/COVID-19 Pandemic"

_diagnostics, 2024, doi:10.3390/diagnostics14202269_

Round 1
Reviewer 1 Report
Comments and Suggestions for Authors
Dear Authors,
thank you for your manuscript, dealing with an interesting issue. Please pay attention to the following comments and questions, pertaining to your manuscript:
1. Line 64. The NPS samples were sent in viral transport medium. Was that the same procedure for the NPA samples? Please make this point clear in your text.
2. Line 68. All patients whose samples were received during epidemiological weeks 33-42 of 2021 were included. Please use the same in the abstract (the current version: between August and October 2021 becomes confusing for your audience).
3. Please explain why did you choose the epidemiological weeks 33-42 for your samples collection. Is that the time period in Colombia with the highest incidence of viral respiratory infections?
4. Figure 1. Could you please explain the origin of the 5083 samples presented in figure 1? There is no explanation about that in the text and I strongly suggest that this piece of information should be added in the text or deleted from figure 1.
5. Table 2 and Table 3 have the same title. According to the text, Table 2 shows the sociodemographic variables and comorbidities in patients infected with one or more respiratory viruses other than SARS-CoV-2 and Table 3 shows the same for the SARS-CoV-2 positive patients. Please correct this point accordingly.
Best Regards
Comments on the Quality of English LanguageModerate editing
Author Response
Manuscript ID diagnostics-3192560
point-by-point response
REVIEWER 1
Comment 1: Line 64. The NPS samples were sent in viral transport medium. Was that the same procedure for the NPA samples? Please make this point clear in your text.
Response 1: Thank you for pointing this out. We shall clarify that NPA samples were collected in sterile saline solution by a trained healthcare professional in accordance with the guidelines of Colombia’s National 65 Institute of Health (INS). Therefore, we corrected the phrase in line 64 (now 77) as follows:
“The NPS samples were sent in viral transport medium and the NPA samples were collected in sterile saline solution by a trained healthcare professional”
Comment 2: Line 68. All patients whose samples were received during epidemiological weeks 33-42 of 2021 were included. Please use the same in the abstract (the current version: between August and October 2021 becomes confusing for your audience).
Response 2: Thank you for the observation. We accept your suggestions, and we shall rewrite line 16 in abstract as follows:
“We received 5083 samples between epidemiological weeks 33-42 of 2021 submitted by the District Health Laboratory of Barranquilla and four local health care institutions during COVID-19 surveillance. Among them, we analyzed 101 samples from individuals presenting with influenza-like illness (ILI)”
Also line 82, as follows:
“In this study, 5083 individuals whose samples were received during epidemiological weeks 33–42 of 2021 (from August to October) were included (Figure 1). For the surveillance process, 101 typical symptomatic ILI cases were selected.”
Comment 3: Please explain why did you choose the epidemiological weeks 33-42 for your samples collection. Is that the time period in Colombia with the highest incidence of viral respiratory infections?
Response 3: Thank you for the comment. We shall clarify that according to sentinel surveillance in previous years this timeframe had the highest incidence of respiratory viruses (See reference 13 from the paper). Also, a convenience criterion was chosen following the SARS-CoV-2 low peak of cases in the city that allowed the surveillance of Flu and ORV by our laboratory. We included the following paragraph in the discussion:
Line 206. “According to sentinel surveillance from previous years, the timeframe chosen (weeks 33-42) presented a high incidence of respiratory viruses [13] and also coincided with a low peak of SARS-CoV-2 in our region which facilitated the work in our laboratory to carry out these experiments. ”
Comment 4: Figure 1. Could you please explain the origin of the 5083 samples presented in figure 1? There is no explanation about that in the text and I strongly suggest that this piece of information should be added in the text or deleted from figure 1.
Response 4: Thank you for pointing this out. We accept your suggestions and accordingly we shall clarify that 5083 samples were submitted by all participant institutions within the time frame that was selected for the study. Therefore, we decided to modify lines 82-84 as follows: “In this study, 5083 individuals whose samples were received during epidemiological weeks 33–42 of 2021 (from August to October) were included (Figure 1). For the surveillance process, 101 typical symptomatic ILI cases were selected.”
And lines 87-90, as follows: “Similarly, pre-existing diseases as a predisposing factor were included in the selection profile. Accordingly, SARS-CoV-2 infection was also used for subsequent grouping of the cases, wherein 36 were positive and 65 were negative for the presence of this virus.”
Comment 5: Table 2 and Table 3 have the same title. According to the text, Table 2 shows the sociodemographic variables and comorbidities in patients infected with one or more respiratory viruses other than SARS-CoV-2 and Table 3 shows the same for the SARS-CoV-2 positive patients. Please correct this point accordingly.
Response 5: Thank you for pointing this out. We kindly accept your suggestions, and as such we shall rewrite those names as follows:
Line 132. “Table 2. Sociodemographic variables and comorbidities among ILI patients who were positive for SARS-CoV-2”
Line 134. “Table 3. Sociodemographic variables and comorbidities among ILI patients who were negative for SARS-CoV-2”. Also, table 3 was modified with the addition of “NEGATIVE SARS-CoV-2 DIAGNOSIS” as a label.
Reviewer 2 Report
Comments and Suggestions for Authors
Congratulations to the authors for the article, it is well written.
However, I believe that these conclusions are not relevant without mentioning the measures imposed by the COVID-19 pandemic, knowing that in 2021, national regulations in various states limited the transmission of the virus, this data must be entered in the introduction part.
Also in the part of discussions, it is necessary to add additional elements to highlight the moment imposed by the pandemic and the similar context in other states.
Author Response
Manuscript ID diagnostics-3192560
point-by-point response
REVIEWER 2
Comment 1: However, I believe that these conclusions are not relevant without mentioning the measures imposed by the COVID-19 pandemic, knowing that in 2021, national regulations in various states limited the transmission of the virus, this data must be entered in the introduction part.
Response 1:
At the beginning 2020, the Ministry of Health and Social Protection of Colombia issued Resolution 385:2020 which invoked the Sanitary Emergency status nationwide. Likewise, another resolution was issued later, resolution 666: 2020. According to this and epidemiological data, we included the following paragraph in the introduction:
Line 53. “Colombia, at the beginning of the COVID emergency in 2020, as many other countries in the world, implemented measures to alleviate and mitigate SARS-CoV-2 propagation and alleviate the burden on the population. Measues including handwashing, social distancing and the use of face mask were mandatory in all social interactions proved an effective way of contention for COVID-19 propagation. However, even with these restrictions, from the beginning of pandemics until the end of 2021, worldwide, Colombia ranked 13th based on the number of COVID-19 cases reported and 11th based on the number of deaths (See WHO: https://www.who.int/docs/default-source/coronaviruse/situationreports/20200326-sitrep-66-covid-19.pdf). In this scenario, the attention of ARI consults was directed towards SARS-CoV-2 infection and reports of Flu and ORV was significantly low. During 2021 ILI cases submitted by sentinel surveillance were 0.02% of the number of cases investigated for SARS-CoV-2 (3.440 ILI /12.189.576 COVID)”
Comment 2: Also in the part of discussions, it is necessary to add additional elements to highlight the moment imposed by the pandemic and the similar context in other states.
Response 2: Thank you for pointing this out. We shall clarify that according to sentinel surveillance in previous years this timeframe had the highest incidence of respiratory viruses (See reference 13 from the paper). Also, a convenience criterion was chosen following the SARS-CoV-2 low peak of cases in the city that allowed the surveillance of Flu and ORV by our laboratory. We included the following paragraph in the discussion:
Line 206. “According to sentinel surveillance from previous years, the timeframe chosen (weeks 33-42) presented a high incidence of respiratory viruses [13] but also coincided with a low peak of SARS-CoV-2 in our region which facilitated the work in our laboratory to carry out these experiments.”
Round 2
Reviewer 1 Report
Comments and Suggestions for Authors
Dear Authors,
thank you for providing comprehensive and convincing answers to my questions and queries and made changes, that have contributed to the optimization of your manuscript and increased the publishing potential of your work. I have no further questions, pertaining to your manuscript.
Best Regards